# Knowledge-Aware Graph Self-Supervised Learning for Recommendation

Shanshan Li [1,2] , Yutong Jia [1,2], You Wu [1,2], Ning Wei [1,2], Liyan Zhang [1,2] and Jingfeng Guo [1,2,*]

1   College of Information Science and Engineering, Yanshan University, Qinhuangdao 066004, China; ssli@stumail.ysu.edu.cn (S.L.); jiayutong@stumail.ysu.edu.cn (Y.J.); youwu@stumail.ysu.edu.cn (Y.W.); weining1121@stumail.ysu.edu.cn (N.W.); zhangly@stumail.ysu.edu.cn (L.Z.)
2   Key Laboratory for Computer Virtual Technology and System Integration of Hebei Province, Qinhuangdao 066004, China
*   Correspondence: jfguo@ysu.edu.cn

**Abstract:** Collaborative filtering (CF) based on graph neural networks (GNN) can capture higher-order relationships between nodes, which in turn improves recommendation performance. Although effective, GNN-based methods still face the challenges of sparsity and noise in real scenarios. In recent years, researchers have introduced graph self-supervised learning (SSL) techniques into CF to alleviate the sparse supervision problem. The technique first augments the data to obtain contrastive views and then utilizes the mutual information maximization to provide self-supervised signals for the contrastive views. However, the existing approaches based on graph self-supervised signals still face the following challenges: (i) Most of the works fail to effectively mine and exploit the supervised information from the item knowledge graph, resulting in suboptimal performance. (ii) Existing data augmentation methods are unable to fully exploit the potential of contrastive learning, because they primarily focus on the contrastive view of data structure changes and neglect the adjacent relationship among users and items. To address these issues, we propose a novel self-supervised learning approach, namely Knowledge-aware Graph Self-supervised Learning (KGSL). Specifically, we calculate node similarity based on semantic relations between items in the knowledge graph to generate a semantic-based item similarity graph. Then, the self-supervised learning contrast views are generated from both the user–item interaction graph and the item similarity graph, respectively. Maximization of the information from these contrastive views provides additional self-supervised signals to enhance the node representation capacity. Finally, we establish a joint training strategy for the self-supervised learning task and the recommendation task to further optimize the learning process of KGSL. Extensive comparative experiments as well as ablation experiments are conducted on three real-world datasets to verify the effectiveness of KGSL.

**Keywords:** self-supervised learning; knowledge graph; semantic similarity; recommendation

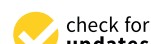



## 1. Introduction

With the development of the Internet, human society has entered an era of information explosion. As an important technological approach to alleviate information overload, recommendation systems have become an indispensable component in many online applications, ranging from E-commerce platforms [1], video-sharing websites [2] to online advertising [3] and so on. Recommender systems aim to mine user preferences based on their historical behavior and provide them with potentially interesting items. At its core, the systems involve studying ways to effectively learn high-quality user and item representation from user historical behavior data [4–6]. Collaborative filtering (CF), as a traditional and effective recommendation method, predicts user preference based on observed user–item interaction behavior. Its fundamental principle is that users with similar interaction behaviors may have similar interests in items [7].

Up to now, various recommendation methods based on CF have been proposed for projecting users and items into latent embedding spaces, such as matrix factorization-based approaches [8–10], autoencoder-based methods [11,12] and graph convolutional network (GCN)-based approaches [13,14]. Due to the large number of network structures naturally possessing recommendation problems, the modeling of high-order connectivity between users and items with graph neural networks (GNN) has become the state-of-art framework in the field of graph learning; examples include GCMC [15], GCCF [16]. However, in real-world scenarios, even when complex user–item interactions are modeled, most CF-based recommendation methods still face the issues of sparse user interaction data and noisy data. For instance, user interactions with items do not necessarily indicate user preference, which might result in hot items. Conversely, the absence of interactions with items does not necessarily imply user disinterest; it might be due to the user's lack of awareness. As a result, it becomes challenging to effectively model the representation of users and items. To address this issue, knowledge graphs (KG), as important external information, have been incorporated into recommendation system (RS) to enhance the representation process of users and items by encoding semantic relatedness.

Existing KG-enhanced recommendation methods are broadly classified into three categories. Firstly, some studies integrate KG embedding with user–item interaction modeling and generate prior item embedding by employing transition-based entity embedding approaches (e.g., TransE [17], TransR [18], etc.). Secondly, to enhance the RS performance in capturing the higher-order semantic information from the KG, some path-based models aim to construct path-oriented user–item connections and incorporate entity information within the KG [19,20]. However, the majority of path-based methods involve the design of a meta-path to generate entity-related relationships, which requires specific domain knowledge and intensive labor to accurately construct paths. Lastly, inspired by the advantages of GNN, recursive information propagation aggregation among multiple nodes while capturing the structural information of graph textcolor[RGB] has become an extremely promising research direction, including as KGAT [21], MIVN [22], KHGT [23], and KGIN [24] approaches. In some scenarios, the recommendation methods integrating KG have achieved some success. However, many of these methods belong to supervised learning methods; model performance heavily relies on high-quality KG (labeled information) and is susceptible to noise disruptions. In real scenarios, knowledge graphs are usually sparse and noisy, and entities have long-tailed distributions which make it challenging to provide accurate and sufficient supervision signals for the model. Consequently, these issues can hinder the generation of accurate user and item representation.

Although supervised learning has achieved success in recommendation systems, it still faces challenges due to the limited availability of training labels, especially considering that labels are often sparse in practical recommendation applications. Recently, the emergence of self-supervised learning (SSL) has provided a new approach for addressing sparse supervision [25]. It learns discriminative embedding from unlabeled data by minimizing the distance between positive samples and maximizing the distance between negative samples. Furthermore, graph neural networks (GNNs), as one of the state-of-the-art machine learning methods, iteratively aggregate neighboring node information to update node representations, effectively capturing both structural information and semantic relationships between nodes [26].

In view of the above-mentioned issues and challenges, we propose a general framework that integrates user–item interaction and knowledge-aware contrastive learning (CL). Firstly, to alleviate the sparsity of user–item interaction data, we introduce the KG to enrich item representation. Subsequently, to deal with the noisy data in both the user–item graph and the knowledge graph, we design a cross-view contrastive learning mechanism aiming to maximize the consistency of nodes across different views, which in turn provides self-supervised signals for learning distinctive node presentation. Finally, a joint optimization strategy is established by combining self-supervised learning tasks with the recommenda-

tion task. This strategy aims to provide personalized recommendations for users. The main contributions of this paper are summarized as follows:

1.  In this paper, we propose a general self-supervised learning paradigm from a novel perspective, which jointly models KG and the user–item interaction graph to improve the robustness of recommendations and alleviate data noise and sparsity problems.
2.  We propose the Knowledge-aware Graph Self-supervised Learning (KGSL) framework, which constructs a contrastive view from the user–item interaction graph and the semantic-based item similarity graph for data augmentation while taking into account both structural information and semantic neighbor information.
3.  Extensive experiments on three real-world datasets are conducted, demonstrating that the proposed KGSL method outperforms several competitive baseline methods. Additionally, ablation studies and parameter investigations are performed to illustrate the impact of unique structures or parameters on model performance.

## 2. Related Work

### 2.1. GNN-Based Recommendation

In recent years, graphs, as data with spatial structures that intuitively describe the relationships between entities in the real world, have attracted widespread attention in both academia and industry [27,28]. In the field of RS, network structure naturally exists among data. To a certain extent, GNN-based models alleviate the data sparsity by aggregating the information of high-order neighbor nodes through information propagation mechanisms [29]. Early recommendation methods, which were based on meta-paths [30] and random walks [31] to generate sequences, used similarity to characterize the semantic relationships among nodes and achieved effective recommendation results. However, such methods heavily relied on substantial manual effort and domain-specific knowledge. Recently, with the success of GNN, a series of graph-based models have been extensively researched in various recommendation scenarios [32]. The GNN-based CF methods incorporate multi-hop connections between users and items into their representation through neighborhood aggregation and node updates. This methods effectively alleviate the impact of data sparsity and improve the performance of recommendation models [32]. For instance, Berg et al. [15] introduced a graph autoencoder into the learning of interaction graphs for the first time to generate the embedding of users and items, which integrated a bilinear decoder to address the rating prediction task in recommendation. Hamilton et al. [33] proposed a GNN-based model named GraphSage, which randomly sampled the neighboring nodes based on the graph topological structure and then aggregated information from the neighboring nodes by using aggregation functions to generate the embedding of central nodes. Subsequently, Stanford University and Pinterest collaborated to propose Pinsage [34], the first industrial-grade GNN recommendation model based on GraphSage, which reduced the computational complexity of GNN models by quickly sampling neighbor nodes using short random walk. In addition, from the perspective of model interpretability, corresponding counterfactual data were designed for different backgrounds, providing a reasonable explanation for the recommended model [35,36]. For model training, in works [37,38], corresponding negative sampling strategies were designed to enhance the robustness of the model. These examples highlight the importance of graph neural networks in the field of recommendation.

### 2.2. Auxiliary Information-Based Recommendation

To mitigate the issue of data sparsity, many studies have incorporated various forms of auxiliary information into the recommendation models. For instance, Ma et al. [39] further learned similar interests between users by jointly factorizing the user social matrices and the user–item interaction matrices, which leveraged social information to alleviate the problem of insufficient interaction data. Wang et al. [40] proposed the collaborative deep learning model that jointly performed deep learning from item content information and collaborative rating information to mitigate the impact of data sparsity. Zhang et al. [41] in-

troduced structured semantic information from the knowledge graph into item embedding using Knowledge Graph Embedding (KGE) algorithms. Gao et al. [42] comprehensively considered the structure, text, and label information of items and used two dual neural networks to learn more accurate item embedding. Zhao et al. [43] transformed the user–item interaction graph into two isomorphic graphs using multiple heterogeneous auxiliary information. They employed two GCNs to learn user and item embedding, which effectively fused auxiliary information and collaborative information to improve the performance of recommendation. Yan et al. [44] proposed a GNN-based recommendation algorithm that combined semantic information with attention, which leveraged the implicit semantic information in text and the influence of interaction relationships in the network to learn the embedding of users and items so as to enhance the accuracy of recommendation results.

*2.3. SSL-Based Recommendation*

Self-supervised learning (SSL) is a learning paradigm that originally emerged in computer vision (CV) [45] and natural language processing (NLP) [46]. Recently, some works have focused on applying SSL for graph representation learning, aiming to explore self-supervised signals by exploring the graph structure. Currently, as one of the leading methods in self-supervised graph representation learning, contrastive learning explores self-supervised signals by comparing multiple contrasting views generated from the same graph, which helps alleviate the 251,188,51 issues of sparse data in recommendation scenarios. Given the sparsity characteristics of most recommendation datasets, researchers have introduced SSL methods into GNN-based models. For example, Zhou et al. [47] designed four self-supervised optimization objectives to learn the correlations in the context information of user–item interaction sequences, and enhanced the data embedding through SSL pre-training. Ma et al. [48] used SSL to predict the users' long-term interaction intentions in the implicit space and generated the interaction subsequence as self-supervised signals for model training. Xia et al. [49] employed SSL to enhance the ability of a hypergraph convolutional network to model the hyper-pair relationships between items in a session, which in turn completed the recommendation by fusing the session representation with the hyper-pair relationships. Wu et al. [50] performed various data augmentation techniques on a user–item graph (node dropout, edge dropout and random walk), which generated the the sub-views of the original graph and constructed an SSL task by maximizing the mutual information between these views. In the joint optimization process with the recommendation task, this approach improved the accuracy of recommendations. Sun et al. [51] designed a hybrid structure of a knowledge graph and a user–item graph to explore self-supervised contrastive learning by generating different data augmentation views. Yang et al. [52] designed a new generative task in the form of masking–reconstructing by calculating rational scores for knowledge triplets, aiming to generate a recommender model with noise-resistant performance.

Although SSL-based GNN recommendation models have been proven effective, the generation of contrastive views often relies on structural perturbation, which can easily disrupt the fundamental essence of the graph. Furthermore, graph self-supervised recommendation models rarely incorporate item KG information, overlooking the rich semantic relationships between items. In response to this, we propose a Knowledge-aware Graph Self-supervised Learning (KGSL) paradigm.

### 3. Preliminaries

*3.1. Notation and Description of Concepts*

KGSL is a GNN-based recommendation algorithm, so the data used in the recommendation algorithm are primarily in the form of graph structure, as shown in Figure 1. They mainly consists of the user–item graph $\mathcal{G}_r$ generated from user–item interaction data, the item knowledge graph $\mathcal{G}_k$ composed of items and their attributes, and the semantic-based item similarity graph $\mathcal{G}_s$ generated from the item knowledge graph. The conceptual de-

scriptions are as follows. Additionally, Table 1 summarizes the parameters used in this paper and their explanations.

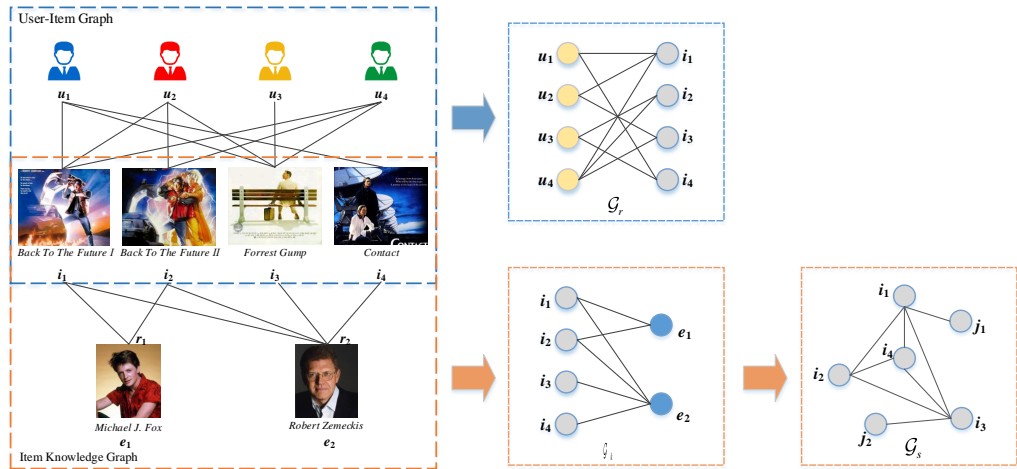

**Figure 1.** Graph Structured data used in KGSL.

**Table 1.** Main parameters and variables in KGSL.

| Parameters | Definitions | Parameters | Definitions |
|---|---|---|---|
| $U$ | User Set | $\mathbf{M}_{rh}, \mathbf{M}_{rt}$ | Mapping Matrix of head and tail entity in KGE |
| $I$ | Item Set | $\mathbf{h}_\perp, \mathbf{h}_\perp$ | Embedding of head and tail entity in $r$ space of KG |
| $\mathbf{A}$ | User–Item Interaction Matrix | $\mathbf{v}_i, \mathbf{v}_j$ | Embedding of item $i, j$ in KG |
| $E$ | Entity Set | $sim(i, j)$ | Similarity of item $i, j$ |
| $R$ | Relation Set | $\rho$ | Rate of edge dropout |
| $\mathcal{G}_r$ | User–Item Graph | $N_r, N_s$ | Node set in graph $\mathcal{G}_r$ and $\mathcal{G}_s$ |
| $\mathcal{G}_k$ | Item Knowledge Graph | $E_r, E_s$ | Edge set in graph $\mathcal{G}_r$ and $\mathcal{G}_s$ |
| $\mathcal{G}_s$ | Semantic-based Item Similarity Graph | $d$ | Embedding dimension |
| $\mathbf{e}_u, \mathbf{e}_i$ | Embedding of User and Item in $\mathcal{G}_r$ | $L$ | Encoder layer |
| $\mathbf{e}_i^s$ | Embedding of Item in $\mathcal{G}_s$ | $K$ | The number of supervised signals |

User–Item Graph: It is denoted as $\mathcal{G}_r = \{(u, y_{ui}, i)|u \in U, i \in I, y_{ui} \in \mathbf{A}\}$, where $U = \{u\}$ denotes the user set, $I = \{i\}$ represents the item set, $\mathbf{A} = \{0,1\}^{|U| \times |I|}$ is the adjacent matrix. If there is an interaction (e.g., click, review, querying) between user $u$ and item $i$, then $y_{ui} = 1$. Otherwise, $y_{ui} = 0$.

Item Knowledge Graph: It is denoted as $\mathcal{G}_k = \{(h, r, t)|h, r \in E, r \in R\}$, where $(h, r, t)$ denotes the triple in the KG, $h, t$ are the head entity and the tail entity, respectively. $r$ is the edge between the head entity and the tail entity, which indicates the relation between them. $E, R$ are the entity set and the relation set, respectively. The head entity and tail entity are connected through different relationships, such as (*The Godfather, is acted by, AI Pacino*), which reflects the fact that the movie *The Godfather* is acted by actor *AI Pacino*.

Semantic-based Item Similarity Graph: It is denoted as $\mathcal{G}_s = \{(i, j)|i, j \in E_s \subset E\}$, where $i, j$ indicates the item nodes in $\mathcal{G}_s$, $(i, j)$ indicates the strong semantic correlation between item $i$ and item $j$, $E_s$ is the subset of entity set $E$ in $\mathcal{G}_k$, which indicates the entity set of $\mathcal{G}_s$.

### 3.2. Self-Supervised Learning

Contrastive learning in SSL is usually used as an auxiliary task to assist the recommendation task by learning additional self-supervised signals. Contrastive SSL aims to learn sample features by comparing similarities and differences among the feature embedding of

different data samples. One of the typical methods to achieve this goal is setting a Noise Contrastive Estimation (NCE) [53]. It is defined by Equation (1).

$$L = \mathbb{E}_{x,x^+,x^-} \left[ -\log \frac{e^{f(x)^{\mathrm{T}}f(x^+)}}{e^{f(x)^{\mathrm{T}}f(x^+)} + e^{f(x)^{\mathrm{T}}f(x^-)}} \right] \qquad (1)$$

where $x$ is similar to $x^+$ but dissimilar to $x^-$, $f(\cdot)$ indicates the encoder, and the encoder functions and similarity measures are designed to vary across different tasks. On the basis of NCE, the existing works further propose Information-theoretic Negative Contrastive Estimation (InfoNCE) [54] that handles dissimilar data pairs, which is defined by Equation (2).

$$L = \mathbb{E}_{x,x^+,x^k} \left[ -\log \frac{e^{f(x)^{\mathrm{T}}f(x^+)}}{e^{f(x)^{\mathrm{T}}f(x^+)} + \sum_{k=1}^{K} e^{f(x)^{\mathrm{T}}f(x^-)}} \right] \qquad (2)$$

In InfoNCE, each sample $x$ corresponds to a set of negative samples $x^k$. By maximizing the mutual information between the positive samples pairs $(x, x^+)$ and minimizing the mutual information between negative samples pairs $(x, x^-)$, the model can better learn more discriminative node embedding. Therefore, when using InfoNCE as the loss function for contrastive self-supervised learning, it is necessary to consider how to construct appropriate contrastive views. This is one of the research focuses of contrastive SSL applied to recommendation tasks.

## 4. The Proposed Methodology

In this section, we provide an overview of the proposed KGSL model, as shown in Figure 2. Firstly, we introduce the node representation learning that incorporates the relational knowledge, which learned from item KG $\mathcal{G}_k$, and generate the semantic-based item similarity graph $\mathcal{G}_s$ by calculating the similarity between nodes. Secondly, the user–item interaction graph $\mathcal{G}_r$ and the semantic-based item similarity graph $\mathcal{G}_s$ are used as the inputs to the KGSL model. The GNN-based encoder is employed to learn node embedding, including user embedding $\mathbf{e}_u$ and item embedding $\mathbf{e}_i$ from the user–item interaction graph $\mathcal{G}_r$, as well as item embedding $\mathbf{e}_i^s$ from the semantic-based item similarity graph $\mathcal{G}_s$. Subsequently, a self-supervised learning task is constructed. By maximizing the consistency of nodes across different views, self-supervision signals are generated to learn discriminative node representation. Finally, the self-supervised learning task and the recommendation task are jointly optimized to provide personalized recommendations for users.

### 4.1. Semantic-Based Item Similarity Graph

The semantic-based item similarity graph $\mathcal{G}_s$ is used as an input to the model, which alleviates the sparsity and noise problems of user–item interaction data. It is generated from item knowledge graph $\mathcal{G}_k$ based on node similarity. The details are described below.

#### 4.1.1. Relationship-Aware Knowledge Aggregation

In item knowledge graph $\mathcal{G}_k$, the item entity is represented as a head entity $h$ and the item entity is represented as a tail entity $t$. The TransD [55] method is employed to learn triples $(h, r, t)$ in the item knowledge graph $\mathcal{G}_k$, aiming to model the semantic relationship between item entities and attribute entities, as illustrated in Figure 3. Firstly, head entity $h$ and tail entity $t$ are projected into the semantic space of relationship $r$, respectively. This projection results in mapping matrices that are related to both entities and relations, which helps distinguish the head and tail entities as different entity types. For instance, in a movie knowledge graph, for the triple combination (movie, directed_by, director_name), the head

and tail entities have different node types. Projection matrices for head entity $h$ and tail entity $t$ are defined as in Equations (3) and (4), respectively.

$$\mathbf{M}_{rh} = \mathbf{r}_p \mathbf{h}_p^{\mathrm{T}} + \mathbf{I} \tag{3}$$

$$\mathbf{M}_{rt} = \mathbf{r}_p \mathbf{t}_p^{\mathrm{T}} + \mathbf{I} \tag{4}$$

where $\mathbf{M}_{rh}$ and $\mathbf{M}_{rt}$ are projection matrices from head entity $h$ and tail entity $t$ in the entity space to the semantic space of $r$, respectively. $\mathbf{h}_p, \mathbf{t}_p, \mathbf{r}_p$ are projection vectors, and $\mathbf{I}$ represents the identity matrix.

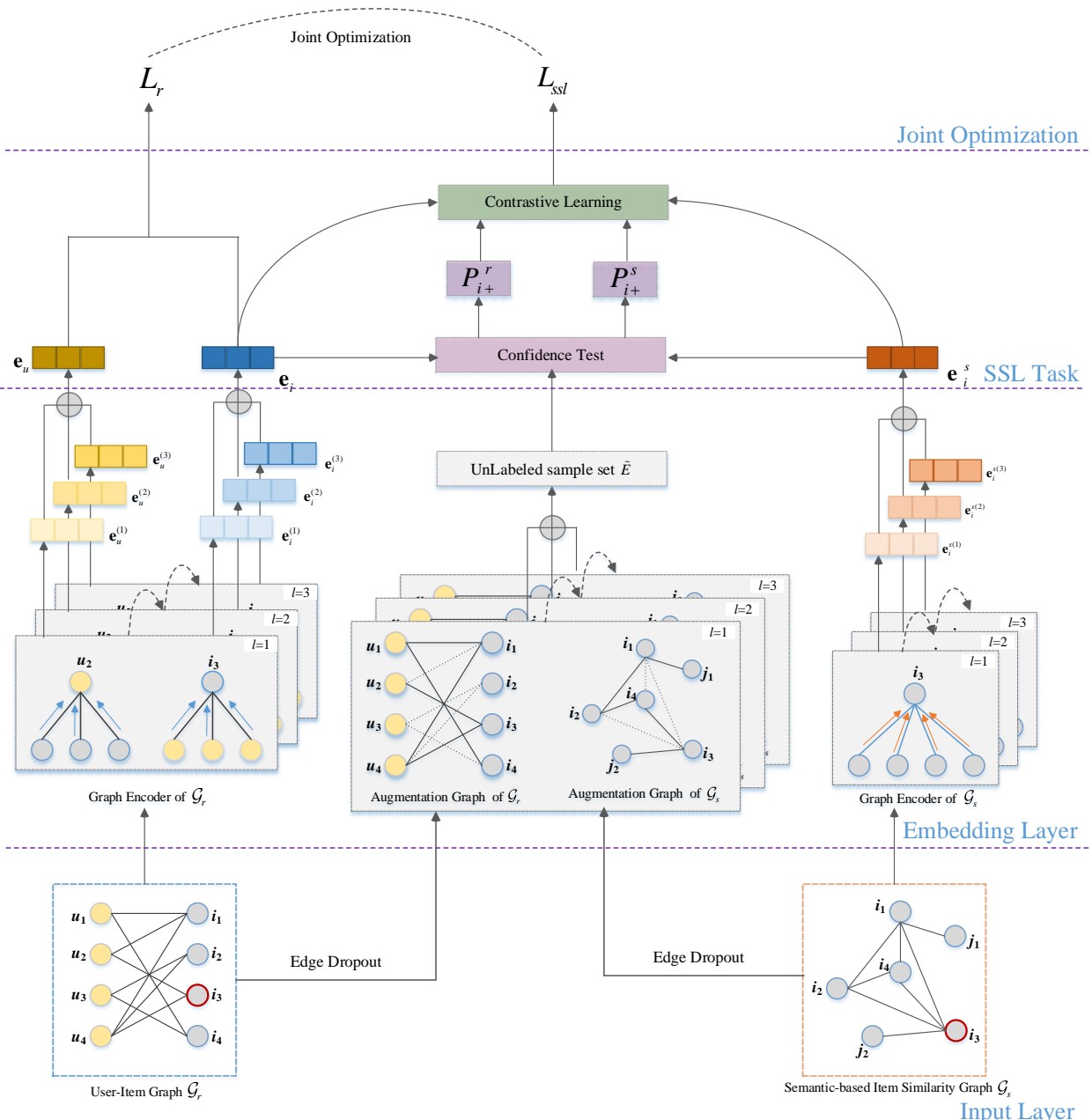

**Figure 2.** The overall framework of KGSL.

Secondly, utilizing the projection matrices, we obtain the embedding of head entity $h$ and tail entity $t$ in the semantic space of relation $r$, denoted as $\mathbf{h}_\perp$ and $\mathbf{t}_\perp$, respectively, as shown in Equations (5) and (6).

$$\mathbf{h}_\perp = \mathbf{M}_{rh}\mathbf{h} \tag{5}$$

$$\mathbf{t}_\perp = \mathbf{M}_{rt}\mathbf{t} \tag{6}$$

Next, the distance between the vectorized representations of head entities and tail entities is measured by the distance function, aiming to measure whether there are correlations between the head and tail entities. The distance scoring function is calculated by Equation (7).

$$f_r(h,t) = -\|\mathbf{h}_\perp + \mathbf{r} - \mathbf{t}_\perp\|_2^2 \tag{7}$$

Finally, in order to learn the node representation that incorporates the item attribute information connected by different relationships, the loss function, which is calculated by Equation (8), is used to train the whole item knowledge graph $\mathcal{G}_k$.

$$L_k = \sum_{(h,r,t)\in\mathcal{G}_k} \sum_{(h',r,t')\notin\mathcal{G}_k} \left[ f_r(h,t) - f_r(h',t') + \gamma \right]_+ \tag{8}$$

where $[x]_+ \triangleq \max(0, x)$, $(h, r, t)$ represents the golden triples that exist in item knowledge graph $\mathcal{G}_k$, while $(h', r, t')$ represents the corrupt triples that are constructed by randomly replacing the head entity or the tail entity, and $\gamma$ represents the minimum distance between positive and negative triples.

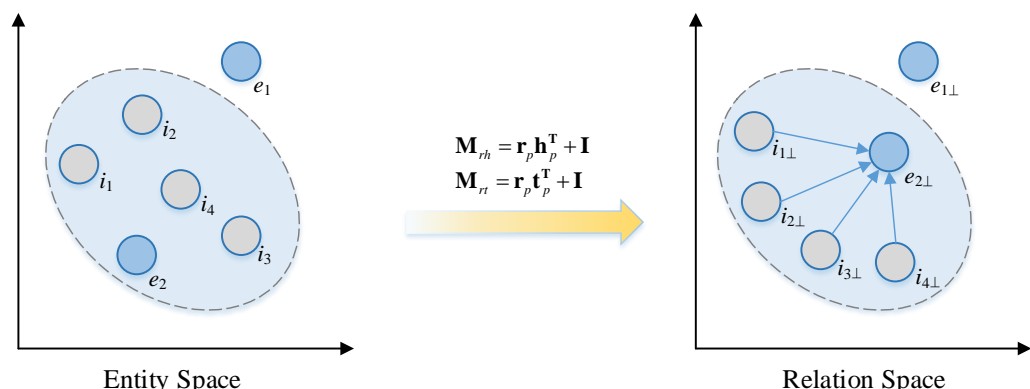

**Figure 3.** Schematic mapping from the entity space into the relationship space.

### 4.1.2. Generating the Semantic-Based Item Similarity Graph

From the above section, the relationship-aware item node embedding is obtained. Next, the semantic similarity between any two item entities is computed by Equation (9), which provides the basis for generating semantic-based item similarity graph $\mathcal{G}_s$.

$$sim(i,j) = \frac{1}{1 + d(\mathbf{v}_i, \mathbf{v}_j)} \tag{9}$$

where $\mathbf{v}_j, \mathbf{v}_j$ stand for vector representations of item $i, j$, which is obtained after pre-training of item knowledge graph $\mathcal{G}_k$, $d(\mathbf{v}_i, \mathbf{v}_i)$ denotes the Euclidean distance between the two item nodes, and $sim(i,j)$ denotes the semantic similarity between item $i$ and item $j$.

Next, based on Equation (9), the similarity between any two nodes is calculated. For each item existing in the user–item interaction graph $\mathcal{G}_r$, we select Top-$k_s$ items with the highest semantic similarity and establish connections to forming semantic-based item similarity graph $\mathcal{G}_s$, as shown in Figure 4.

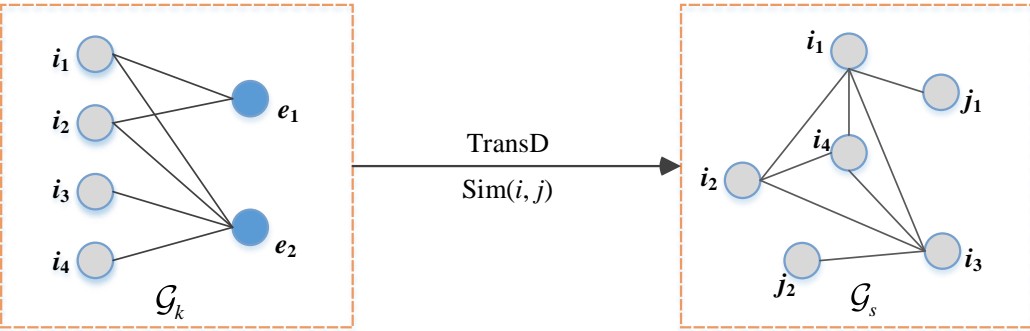

**Figure 4.** The generation process of semantic-based item similarity graph $\mathcal{G}_s$.

Through the above approach, the TransD method is used to learn the node representations from item knowledge graph $\mathcal{G}_k$ that contains rich semantic information. Node similarity is calculated based on distance Equation (9), and a fixed number of neighboring nodes is generated for each node. This effectively mitigates the impact of noisy information from weakly relevant nodes during the subsequent information aggregation process in the model.

### 4.2. Representation Learning Based on GNN

GNN-based approaches generate the representations for users and items by propagating and aggregating neighborhood information on graph-structured data. In KGSL, GNNs are employed to model user–item interaction graph $\mathcal{G}_r$ and semantic-based item similarity graph $\mathcal{G}_s$. This results in learning the embedding for users $\mathbf{e}_u$, collaborative embedding for items, $\mathbf{e}_i$, and semantic embedding for items $\mathbf{e}_i^s$. Specifically, following the principles of LightGCN, the non-linear activation and feature transformation in the propagation function are discarded. Formally, this is calculated by Equations (10)–(12).

$$\mathbf{e}_u^{(l)} = \sum_{i \in N_u} \frac{1}{\sqrt{|N_u|}\sqrt{|N_i|}} \mathbf{e}_i^{(l-1)} \tag{10}$$

$$\mathbf{e}_i^{(l)} = \sum_{u \in N_i} \frac{1}{\sqrt{|N_i|}\sqrt{|N_u|}} \mathbf{e}_u^{(l-1)} \tag{11}$$

$$\mathbf{e}_i^{s(l)} = \sum_{j \in N_i^s} \frac{1}{\sqrt{|N_j^s|}\sqrt{|N_i^s|}} \mathbf{e}_j^{s(l-1)} \tag{12}$$

In Equations (10)–(12), $\mathbf{e}_u^{(l)}$ and $\mathbf{e}_i^{(l)}$ denote the $l$-th layer representation of user $u$ and item $i$ in user–item interaction graph $\mathcal{G}_r$, respectively. $N_u$ denotes the item sets that have interacted with user $u$, while $N_i$ denotes the set of users who have interacted with item $i$. In Equation (12), $\mathbf{e}_i^{s(l)}$ denotes the $l$th layer representation of item $i$ in the semantic-based item similarity graph $\mathcal{G}_s$, and $N_i^s$ represents the first-order neighbor set of item $i$ in the semantic-based item similarity graph $\mathcal{G}_s$.

After propagation of $L$ layers, each layer of embedding contains the information aggregated from the previous layer neighboring nodes. This achieves the effect of modeling the higher-order connectivity information between users and items on graph $\mathcal{G}_r$ and expanding the semantic similarity of the items on graph $\mathcal{G}_s$. Here, a weighted summation readout

function is used to combine the node representations obtained from each layer, and the result is taken as the final representation of the node, as shown in Equation (13).

$$\mathbf{e}_u = \frac{1}{L+1} \sum_{l=0}^{L} \mathbf{e}_u^{(l)}$$

$$\mathbf{e}_i = \frac{1}{L+1} \sum_{l=0}^{L} \mathbf{e}_i^{(l)} \tag{13}$$

$$\mathbf{e}_i^{\mathfrak{s}} = \frac{1}{L+1} \sum_{l=0}^{L} \mathbf{e}_i^{\mathfrak{s}(l)}$$

When $l = 0$, it represents the corresponding initial input; when $l \in [1, L]$, it stands for the node representation obtained after aggregation through the $l$ layer of neural networks.

*4.3. Construction of the Self-Supervised Learning Task*

After obtaining item embedding $\mathbf{e}_i$ that contains user preference on user–item interaction graph $\mathcal{G}_r$ as well as item embedding $\mathbf{e}_i^s$ on semantic-based item similarity graph $\mathcal{G}_s$, a self-supervised learning task is constructed for both. KGSL constructs the self-supervised learning task aiming at learning the common features of similar nodes in different views and distinguishing the differences between dissimilar entities. The purpose is the generation of additional self-supervised signals to alleviate the sparsity of the interaction data and to eliminate noisy data to enhance the robustness of the model.

Firstly, in order to extract additional self-supervised signals from the raw data, data augmentation is performed on both user–item interaction graph $\mathcal{G}_r$ and semantic-based item similarity graph $\mathcal{G}_s$. Specifically, during each iteration of GNN, edge dropout operations are applied to both graph $\mathcal{G}_r$ and graph $\mathcal{G}_s$ with a certain probability $\rho$. In this way, the data augmentation views for both are constructed and the unlabeled sample set is created, where $\rho$ is a trainable hyper-parameter. This approach is used to make it more conducive for the model to identify influential nodes in the views, reducing the sensitivity of node representation to structural changes. This is formalized in Equations (14) and (15).

$$\widetilde{\mathcal{G}}_r = (N_r, \mathbf{M}_r \odot E_r) \tag{14}$$

$$\widetilde{\mathcal{G}}_s = (N_s, \mathbf{M}_s \odot E_s) \tag{15}$$

In Equations (14) and (15), $N_r$ denotes the node set of graph $\mathcal{G}_r$, which includes all user nodes and item nodes. $N_s$ represents the set of nodes, which includes all item nodes in semantic-based item similarity graph $\mathcal{G}_s$. $E_r$ and $E_s$ are the edge sets in $\mathcal{G}_r$ and $\mathcal{G}_s$, respectively. $\mathbf{M}_r \in \{0,1\}^{|E_r|}$ and $\mathbf{M}_s \in \{0,1\}^{|E_s|}$ are two masking vectors used to randomly drop out edges in $\mathcal{G}_r$ and $\mathcal{G}_s$. $\widetilde{\mathcal{G}}_r$ and $\widetilde{\mathcal{G}}_s$ represent the views after performing edge dropout operations on $\mathcal{G}_r$ and $\mathcal{G}_s$, which are the data augmentation views. Additionally, during training, an extra graph encoder is utilized to learn the representation of item nodes in the augmented views, denoted as the unlabeled sample set $\widetilde{E}$. The graph encoder structure is the same as in Section 4.2 and is not further elaborated in this section.

Collaborative embedding $\mathbf{e}_i$ and semantic embedding $\mathbf{e}_i^s$ of item $i$ reflect different aspects of item information, and they can seek self-supervised signals from each other. Taking the example of predicting self-supervised signals for user–item interaction graph $\mathcal{G}_r$ based on semantic item similarity graph $\mathcal{G}_s$, self-supervised signals are predicted from the unlabeled sample set $\widetilde{E}$ that incorporates auxiliary information and is beneficial for the recommendation task. Specifically, for any given item $i$ in the user–item interaction graph $\mathcal{G}_r$, a confidence test is performed on the items in the unlabeled sample set $\widetilde{E}$. This involves calculating probability $y_{i+}^r$ that items in unlabeled sample set $\widetilde{E}$ exhibit a positive semantic with respect to the given item $i$, as shown in Equation (16).

$$y_{i+}^r = \text{Softmax}(\cos(\widetilde{\mathbf{e}}, \mathbf{e}_i^s)) \tag{16}$$

where $\widetilde{\mathbf{e}}$ denotes item embedding in the unlabeled sample set $\widetilde{E}$, and $\cos(\cdot)$ represents the cosine similarity. $y^r_{i+}$ represents the additional supervised signals of user–item interaction graph $\mathcal{G}_r$, which is obtained by computing the similarity between the items in semantic-based item similarity graph $\mathcal{G}_s$ and the items in unlabeled sample set $\widetilde{E}$.

Then, based on the results of confidence test $y^r_{i+}$, the Top-K unlabeled positive semantic samples with the highest confidence are selected as positive samples for item $i$. These selected samples serve as self-supervised signals for item $i$ in user–item interaction graph $\mathcal{G}_r$. The specific process is described in Equation (17).

$$P^r_{i+} = \{\widetilde{\mathbf{e}}_k \mid k \in \text{Top-K}(y^r_{i+}), \widetilde{E} \sim \widetilde{\mathcal{G}}_r\} \tag{17}$$

where $\widetilde{\mathbf{e}}_k$ represents the Top-K item embedding with the highest confidence after the confidence test, which constitutes the self-supervised signal set $P^r_{i+}$ of item $i$ in user–item interaction graph $\mathcal{G}_r$.

Similarly, self-supervised signal set $P^s_{i+}$, predicted by semantic-based item similarity graph $\mathcal{G}_s$, can be obtained by calculating the similarity between the items in user–item interaction graph $\mathcal{G}_r$ and the items in unlabeled sample set $\widetilde{E}$, as shown in Equations (18) and (19).

$$y^s_{i+} = \text{Softmax}(\cos(\widetilde{\mathbf{e}}, \mathbf{e}^r_i)) \tag{18}$$

$$P^s_{i+} = \{\widetilde{\mathbf{e}}_k \mid k \in \text{Top-K}(y^s_{i+}), \widetilde{E} \sim \widetilde{\mathcal{G}}_s\} \tag{19}$$

During the model training phase, KGSL utilizes the two generated knowledge-aware views to co-supervise each other. Following the existing self-supervised learning paradigm, we take the items that are semantically positive as calculated in Equation (17) and the items that interact with users as positive samples for user–item interaction graph $\mathcal{G}_r$. Other nodes constitute the set of negative samples. To improve computational efficiency, a random sampling strategy is employed to randomly sample hard negative samples from the set of negative samples for model training. The auxiliary supervision signals from positive sample pairs encourage consistency between the same nodes in different views, while the supervision signals from negative sample pairs enhance the differences between different nodes. Formally, we use the InfoNCE loss function to maximize the consistency between positive sample pairs and minimize the consistency between negative sample pairs, as shown in Equation (20).

$$L^r_{ssl} = -\log \frac{\sum_{p \in P^r_{i+}} \exp(\cos(\mathbf{e}_i, \widetilde{\mathbf{e}}_p)/\tau)}{\sum_{p \in P^r_{i+}} \exp(\cos(\mathbf{e}_i, \widetilde{\mathbf{e}}_p)/\tau) + \sum_{j \in I/P^r_{i+}} \exp(\cos(\mathbf{e}_i, \widetilde{\mathbf{e}}_j)/\tau)} \tag{20}$$

In Equation (20), $\mathbf{e}_i$ represents item embedding in user–item interaction graph $\mathcal{G}_r$, $\widetilde{\mathbf{e}}_p$ represents one of the self-supervised signals for $\mathbf{e}_i$, $\widetilde{\mathbf{e}}_j$ represents item embedding from unlabeled sample set $\widetilde{E}$ that is not labeled as a self-supervised signal. $\tau$ is the temperature coefficient of the Softmax function, and an appropriate temperature coefficient allows the model better learning of hard negative samples. Following the existing research that introduces self-supervised learning into the recommendation system, in the training of the KGSL model, temperature coefficient $\tau$ is set to 0.1.

Based on Equation (19), we obtain semantic positive samples of semantic-based item similarity graph $\mathcal{G}_s$, and the positive sample set is generated by combining the first-order neighbors in $\mathcal{G}_s$. Negative sample sets are generated for each node, and the hard negative samples are randomly selected from them to participate in model training. Similarly, loss function $L^s_{ssl}$ with an enhanced self-supervised learning objective is defined by Equation (21).

$$L^s_{ssl} = -\log \frac{\sum_{p \in P^s_{i+}} \exp(\cos(\mathbf{e}^s_i, \widetilde{\mathbf{e}}_p)/\tau)}{\sum_{p \in P^s_{i+}} \exp(\cos(\mathbf{e}^s_i, \widetilde{\mathbf{e}}_p)/\tau) + \sum_{j \in I/P^s_{i+}} \exp(\cos(\mathbf{e}^s_i, \widetilde{\mathbf{e}}_j)/\tau)} \tag{21}$$

Since the denominator of contrastive loss function (Equations (20) and (21)) is the similarity of the node to all its positive and negative sample nodes, and the positive

sample consists of structural and semantic positive samples, this result is unaffected when counterfactual data are present. Combining the two views of the item embedding with the self-supervised signal for mutual information maximization comparison, we obtain the final loss function for the self-supervised learning task, which is calculated by Equation (22).

$$L_{ssl} = L_{ssl}^r + L_{ssl}^s \tag{22}$$

### 4.4. Joint Optimization

To alleviate the challenges posed by the sparse interaction data and to further improve the accuracy of KGSL recommendation, we introduce a joint optimization strategy that combines self-supervised learning tasks with supervised recommendation tasks, that is, creating a mutually reinforcing learning process.

To optimize the recommendation task, KGSL utilizes the classic Bayesian Personalized Ranking Loss (BPR) [10], which is commonly used in Top-K recommendation algorithms. Specifically, it assumes that for a given user, the observed interactions (in this paper, denoting positive samples) indicate greater user preference, and they should be assigned higher prediction values than the unobserved interactions (in this paper, denoting negative samples). It is calculated by Equation (23).

$$L_r = \sum_{(u,i,j) \in O} -\ln \sigma(\hat{y}_{ui} - \hat{y}_{uj}) \tag{23}$$

where $O = \{(u,i,j)|(u,i) \in O^+, (u,j) \in O^-\}$ is the training data of the model $O^+$ denotes the observed user–item interactions (positive samples), and $O^-$ represents the unobserved user-item interactions (negative samples). $\hat{y}_{ui}$ represents the predicted rating of user $u$ for item $i$, and the rating function is defined using inner product operation, denoted as $\hat{y}_{ui} = \hat{\mathbf{e}}_u^{\mathrm{T}} \mathbf{e}_i$.

Finally, the KGSL model jointly optimizes the self-supervised learning task and the recommendation task, which is defined by Equation (24).

$$L_{KGSL} = L_r + \beta L_{ssl} + \lambda \|\Theta\|_2^2 \tag{24}$$

where $\beta$ is the hyper-parameter that regulates the scale of self-supervised learning, $\lambda$ is the hyper-parameter that controls the strength of the regularization, and $\Theta = \{\mathbf{e}_u, \mathbf{e}_i, \mathbf{e}_i^s\}$ are the parameters that the model needs to learn.

The detailed process of the KGSL recommendation algorithm based on item knowledge-aware graph self-supervised learning is shown in Algorithm 1.

### 4.5. Complexity Analysis of KGSL

To optimize the multi-task objective in Equation (24), we decouple the training process into four parts: adjacency matrix normalization, graph convolutional network, self-supervised learning task, and recommendation task. We iteratively update the corresponding parameters to minimize the loss until achieving the best performance on the validation set. To facilitate the analysis of the complexity of each node, $N$ represents the total number of users and items, $|E_r|$ represents the number of edges in the user–item interaction graph $\mathcal{G}_r$, $|E_s|$ represents the number of edges in the semantic-based item similarity graph $\mathcal{G}_s$, $d$ is the embedding dimension, $B$ is the batch size for training, $L$ is the layers of GNN, and $\hat{\rho} = 1 - \rho$ represents the probability that an edge is retained. Next, we explain the time complexity of each part.

- Adjacency Matrix Normalization: Before performing graph convolution operations, it is necessary to normalize the adjacency matrix of the graph. In KGSL, for each training iteration, we need to generate augmented views for both user–item interaction graph $\mathcal{G}_r$ and semantic-based item similarity graph $\mathcal{G}_s$. Since the number of non-zero elements in the original graph and the augmented views are $2(|E_r| + |E_s|)$

and $2\hat{\rho}(|E_r| + |E_s|)$, respectively, the overall computational complexity of this part is $O\big(2(1 + \hat{\rho})(|E_r| + |E_s|)\big)$.

- GNN: For the $l$ th convolutional layer, the complexity of performing matrix multiplication is $O\big(2d(|E_r| + |E_s|)\big)$. Therefore, the complexity of graph convolution with a total number of layers $L$ is $O(2Ld(|E_r| + |E_s|))$. Thus, adding the complexity of performing graph convolutions on two augmented views, the overall complexity becomes $O\big(2Ld(1 + \hat{\rho})(|E_r| + |E_s|)\big)$.

- SSL Task: For calculating the time complexity of the self-supervised tasks, only inner product operations are considered. As shown in Equation (20), when calculating the loss for item nodes in the user–item interaction graph $\mathcal{G}_r$, all other item nodes are treated as a negative sample. Since KGSL sets up two self-supervised tasks, the overall time complexity is denoted as $O\big(4(Bd + B^2 d)\big)$.

- Recommendation Task: Similarly, considering only inner product calculations, the calculation complexity is assessed. Since the BPR method computes the loss function by matching each positive sample with a negative sample, the overall computational complexity for the entire training process is denoted as $O(2Bd)$.

---

**Algorithm 1:** The Algorithm of KGSL

---

**Input:** User–Item Interaction Graph $\mathcal{G}_r$;
    Semantic-based Item Similarity graph $\mathcal{G}_s$;
    User Set $U$; Item Set $I$;
    User Embedding $\mathbf{e}_u$; Item Embedding $\mathbf{e}_i$;
    Temperature Parameter $\tau$;
    Regularization Coefficient $\beta$;
    Train Times $\max_{\text{iter}}$;
    Train Sample $B_{\text{train}}$
**Output:** Top-K recommended items list for the user
 Randomly Initialize user embedding $\mathbf{e}_u$ and item embedding $\mathbf{e}_i$;
**for** iter $= 1, \dots, \max_{\text{iter}}$ **do**
    Perform edge dropout on $\mathcal{G}_r$ and $\mathcal{G}_s$ to generate unlabeled sample set $\widetilde{E}$
    **for** batchsize in $B_{\text{train}}$ **do**
      Construct a self-supervised task and extract self-supervised signals from $\widetilde{E}$
      **for** item $i$ in $I$ **do**
        (**a**) the self-supervised signals of $\mathcal{G}_r$
        Calculate the positive semantic from the unlabeled sample set $\widetilde{E}$
        (Equation (16))
        Calculate the self-supervised signal set for the $i$ in $\mathcal{G}_r$ (Equation (17));
        Calculate the loss function of $\mathcal{G}_r$ (Equation (20))
        (**b**) the self-supervised signals of $\mathcal{G}_s$
        Calculate the positive semantic from the unlabeled sample set $\widetilde{E}$
        (Equation (18))
        Calculate the self-supervised signal set for the $i$ in $\mathcal{G}_s$ (Equation (19));
        Calculate the loss function of $\mathcal{G}_s$ (Equation (21))
      **end**
    **end**
    Calculate the Self-supervised task loss $L_{ssl}$ (Equation (22))
    Calculate the recommendation task loss $L_r$ (Equation (23))
    Update all parameters according to Equation (24) with Adam;
**end**
**return** Top-K recommendation list

---

## 5. Experiment

### 5.1. Experimental Setup

5.1.1. Dataset Description

To validate the effectiveness of the KGSL model, we conduct extensive experiments on three publicly available datasets from different domains. These datasets are different

in size and sparsity; we include MovieLens-1M, Last-FM, and Book-Crossing, which are described in detail below.

- MovieLens-1M, a movie recommendation dataset obtained from the MovieLens website, a movie recommendation service platform. It contains over 1 million explicit ratings from more than 6000 users for over 4000 movies. User ratings for movies range from 1 to 5. In addition to rating information, this dataset also includes some auxiliary information. Microsoft Satori organized movies and their associated attribute entities into a knowledge graph, which is used for research and development in a personalized recommendation system.
- Last-FM, a music recommendation dataset collected from the online music platform Last.fm. It includes the listening history records of over 1000 users on the Last.fm website over the course of a year. This dataset covers more than 4000 artists and over 10,000 songs. Additionally, the dataset includes information about artists, songs, labels, and genres. Microsoft Satori also organized this information into a corresponding knowledge graph.
- Book-Crossing, a book recommendation dataset provided by the social networking site Book-Crossing, which is focused on readers. It includes rating and content information for over 27,000 books available on the website. User ratings for books in this dataset range from 1 to 10. Similar to the previous two datasets, the original dataset book content information is also present in the corresponding knowledge graph created by Microsoft Satori.

Since KGSL is a recommendation algorithm developed based on implicit feedback, three datasets need to be preprocessed. Firstly, the three datasets are converted from explicit ratings to implicit feedback. Positive interaction records of users with items are labeled as 1, indicating that the user has interacted with the item. Negative interaction records, representing the absence of user–item interaction, are labeled as 0. For the MovieLens-1M dataset, the rating threshold for its positive evaluation is set to 5, while for the Last-FM and Book-Crossing datasets, no positive rating threshold is set due to their high sparsity. Secondly, to fully utilize the auxiliary information of the items, head entities of the triples in the knowledge graph and the original dataset scores are retained according to the correspondence between the auxiliary information in the original dataset and the item entities in the knowledge graph. After the aforementioned preprocessing steps, the specific statistics information of the three datasets are as shown in Table 2.

**Table 2.** The statistics of the dataset.

|                | #Users | #Items | #Interactions | #Entities | #Relations | #Triples |
|----------------|--------|--------|---------------|-----------|------------|----------|
| MovieLens-1M   | 5986   | 2347   | 298,856       | 6729      | 8          | 20,195   |
| Last-FM        | 1872   | 3846   | 42,346        | 9366      | 60         | 15,518   |
| Book-Crossing  | 17,860 | 14,910 | 139,746       | 24,039    | 10         | 19,793   |

For the preprocessed datasets, we employ a 5-fold cross-validation approach to train and evaluate the model performance. The datasets are divided into five equal parts. In each iteration, four parts are used for training the model, and the one remaining part is used for validation. This process is repeated five times, and the final experimental results are obtained by averaging the validation results from these five iterations.

### 5.1.2. Evaluation Metrics

KGSL primarily employs Top-K for model evaluation. There are two ways to evaluate the Top-K metric: sampling evaluation and full evaluation. Sampling evaluation involves predicting ratings for a fixed number of non-interaction negative samples for each user. On the other hand, full evaluation predicts ratings for all uninteracted negative samples for each user, ranks all the negative samples, and generates a Top-K recommendation list. Performance metrics are then calculated based on this list. Compared to sampling evalu-

ation, full evaluation provides a more comprehensive and accurate assessment of model performance. To evaluate model performance, we use Hit Ratio (HR), Recall and Normalized Discounted Cumulative Gain (NDCG), which are obtained by Eqautions (25)–(28), respectively.

$$\text{HR} = \sum_{u=1}^{|U|} \frac{\text{hit}(u)}{|U| * K} \tag{25}$$

The denominator consists of the entire test set, while the numerator represents the sum of the number of items from the Top-K list that belong to the test set for each user.

$$\text{Recall} = \frac{\text{TP}}{\text{TP} + \text{FN}} \tag{26}$$

where TP (True Positive) represents an item that was recommended and also appeared in the user interaction list, indicating that the recommended item matches the user interest, and FN (False Negative) represents an item that was not recommended but appeared in the user interaction list, indicating that the model did not accurately recognize the user interest. To calculate the Normalized Discounted Cumulative Gain (NDCG), we first need to calculate the Discounted Cumulative Gain (DCG), as shown in Equation (27).

$$\text{DCG@K} = \sum_{i}^{K} \frac{2^{r(i)} - 1}{\log_2(i+1)} \tag{27}$$

where $r(i)$ represents the relevance between the item located in the $i$th position in the recommendation list and the user interest. Typically, $r(i) = 1$ indicates that the user is interested (i.e., a real interaction exists with the item), and $r(i) = 0$ indicates that the user is not interested (i.e., no interaction exists with the item). Subsequently, dividing the DCG value by the theoretically maximum value of DCG and then normalizing yields NDCG, which is defined as shown in Equation (28).

$$\text{NDCG@K} = \frac{\text{DCG@K}}{\text{IDCG@K}} \tag{28}$$

where IDCG@K represents the Ideal Discounted Cumulative Gain for the best recommendation list predicted by the model for the user. Compared with the HR, Recall and NDCG take into account not only the quantity of correct samples but also their relative positions and relevance in the recommendation list. It is a more comprehensive and reliable metric for evaluating the effectiveness of the recommendation system.

5.1.3. Baselines

In this section, we introduce the baselines compared with the KGSL model. These recommendation algorithm can be categorized into the following three categories: NN-based Recommendation System (NeuMF); GNN-based Recommendation System (NGCF, LightGCN); SSL-based Recommendation System (SGL, MCCLK).

**Neural Networks for Recommendation**

- **NeuMF** [8] is an NN-based CF recommendation algorithm. It employs neural networks instead of matrix factorization to simulate higher-order interactions and learns more complex nonlinear interaction features. In the comparative experiments, the model's entity embedding dimension is set to 50, and the number of layers in the graph encoder is set to 2.

**Graph Neural Network for Recommendation**

- **NGCF** [14] is a GNN-based recommendation algorithm. It organizes user-item interaction data into the form of a user-item interaction bipartite graph. It utilizes the information propagation and aggregation mechanism of GNN to explicitly encode high-order connectivity between users and items into collaborative information. Finally, it uses user and item embedding containing high-order collaborative infor-

mation to make rating prediction. In the comparative experiments, the model's entity embedding dimension is set to 50, and the number of layer in the graph encoder is set to 2.

- **LightGCN** [13] is a GNN-based recommendation algorithm that builds upon the NGCF model. It introduces a lightweight graph convolution operation to learn user–item bipartite graphs. Instead of using non-linear activation functions and feature transformation operations in graph neural networks, LightGCN replaces them with simple weighted aggregators. This further enhances the training efficiency of the recommendation algorithm and the encoding capability of user–item embedding vectors. In the comparative experiments, the model's entity embedding dimension is set to 50, and the number of layers in the graph encoder is set to 2.

**Self-Supervised Learning for Recommender Systems**

- **SGL** [50] is an SSL recommendation algorithm based on graph neural networks. Its SSL task involves data augmentation operations based on graph structure perturbation on the user–item interaction graph. Then, it maximizes mutual information between embedding of the same node under different views. In the comparative experiments, the model's entity embedding dimension is set to 50, and the number of layers in the graph encoder is set to 2.
- **MCCLK** [56] is an SSL recommendation algorithm based on a graph neural network. It takes a user–item interaction graph and an item–entity knowledge graph as separate local views, then concatenates them to generate a user–item–entity graph as the global view. Finally, it designs self-supervised learning tasks based on a multi-level cross-view contrastive learning mechanism to enhance the recommendation task. In the comparative experiments, the model's entity embedding dimension is set to 50, and the number of layers in the graph encoder is set to 2.

*5.2. Performance Comparison with Baselines*

The section aims to evaluate the effectiveness of the KGSL model. The performance of the KGSL model is compared with the baseline approaches on the three datasets using Recall and NDCG as evaluation metrics in both Top-10 and Top-20 recommendation tasks. The experimental results are shown in Table 3. In the table, the best results are indicated in bold, and the sub-optimal results are underlined.

**Table 3.** Performance comparison of different recommendation models on three datasets.

| Dataset | Metric | NeuMF | NGCF | LightGCN | SGL | MCCLK | KGSL | Improve (%) |
|---|---|---|---|---|---|---|---|---|
| MovieLens-1M | Recall@10 | 21.468 | 21.607 | 24.316 | 24.609 | 24.738 | **25.875** | 4.60% |
| | NDCG@10 | 18.734 | 21.223 | 22.097 | 22.733 | 22.931 | **23.834** | 3.94% |
| | HR@10(%) | 9.002 | 10.019 | 10.696 | 6.326 | 7.923 | **11.058** | 3.38% |
| | Recall@20 | 29.818 | 30.556 | 32.916 | 33.470 | 33.835 | **34.852** | 3.01% |
| | NDCG@20 | 23.224 | 24.293 | 26.417 | 27.269 | 27.331 | **27.786** | 1.66% |
| | HR@20(%) | 6.717 | 7.343 | 7.708 | 4.921 | 5.791 | **7.914** | 2.67% |
| Last-FM | Recall@10 | 19.363 | 27.396 | 27.927 | 28.141 | 27.993 | **29.585** | 5.13% |
| | NDCG@10 | 13.944 | 20.635 | 20.760 | 20.948 | 20.901 | **22.181** | 5.89% |
| | HR@10(%) | 3.866 | 4.770 | **5.968** | 4.072 | 4.536 | 5.937 | -0.52% |
| | Recall@20 | 25.641 | 29.865 | 34.814 | 29.994 | 35.903 | **37.362** | 4.06% |
| | NDCG@20 | 14.801 | 16.453 | 22.563 | 19.554 | 23.012 | **24.141** | 4.91% |
| | HR@20(%) | 2.832 | 3.473 | 4.027 | 2.533 | 2.817 | **4.078** | 1.27% |
| Book-Crossing | Recall@10 | 7.701 | 7.915 | 9.263 | 8.904 | 9.535 | **9.952** | 4.37% |
| | NDCG@10 | 4.767 | 4.498 | 5.795 | 5.363 | 6.051 | **6.174** | 2.03% |
| | HR@10(%) | 1.348 | 1.128 | 1.564 | 1.102 | 1.381 | **1.594** | 1.92% |
| | Recall@20 | 10.955 | 9.230 | 11.251 | 10.821 | 11.537 | **12.088** | 4.78% |
| | NDCG@20 | 5.745 | 4.737 | 6.414 | 5.906 | 6.442 | **7.161** | 11.16% |
| | HR@20(%) | 0.896 | 0.783 | 1.028 | 0.689 | 0.937 | **1.054** | 2.73% |

The HR metric is expressed as a percentage value. From the experimental results in Table 3, we have the following observations and analysis: (1) In the Top-10 recommendation task, KGSL outperforms all five compared baseline methods on all three datasets, except for the HR metric on the Last.fm dataset. On the MovieLens-1M dataset, KGSL achieves an improvement of 4.60% in Recall@10 and an improvement of 3.94% in NDCG@10 compared to the suboptimal method MCCLK. In terms of the HR@10 metric, KGSL improves by 3.38% compared to LightGCN. On the Last-FM dataset, KGSL achieves an improvement of 5.13% in Recall@10 and an improvement of 5.89% in NDCG@10 compared to the suboptimal method SGL, while showing a slight decrease of 0.52% in the HR@10 metric compared to the best-performing method. On the Book-Crossing dataset, KGSL achieves an improvement of 4.37% in Recall@10 and an improvement of 2.03% in NDCG@10 compared to the suboptimal method MCCLK, and outperforms LightGCN by 1.92% in the HR@10 metric.

(2) In the Top-20 recommendation task, KGSL also outperforms all five baseline methods on all three datasets. Across the three datasets, KGSL achieves a 3.01%, a 4.06% and a 4.78% improvement in Recall@20 and 1.66%, 4.91% and 11.16% improvement in NDCG@20 compared to the suboptimal method MCCLK. Compared with the second-best LightGCN on HR@20, there are improvements of 2.67%, 1.27% and 2.73%, respectively. These experiments demonstrate that the self-supervised learning task of KGSL is effective in assisting the model to better train the recommendation task. This self-supervised learning task leverages semantic similarity among items to generate effective self-supervised signals, effectively expanding the existing supervised information in model training. This improves the KGSL ability to learn embedding for users and items, resulting in more accurate recommendation.

(3) From Table 3, it is evident that on all three datasets, GNN-based methods (KGSL, MCCLK, SGL, LightGCN, NGCF) outperform the neural network-based method (NeuMF). This demonstrates the superiority of graph neural networks in modeling collaborative information between users and items on the user–item interaction graph compared to modeling user–item interactions using deep neural networks.

(4) In the Top-10 recommendation task, the self-supervised models, SGL and MCCLK, achieved higher Recall values and NDCG values on the three data sets, but the values were slightly lower than those of the neural network model LightGCN in HR indicators. In the Top-20 recommendation task, SGL outperformed LightGCN only on the MovieLens-1M dataset, while MCCLK performed better than LightGCN on all three datasets, except for the HR metric. In both recommendation tasks, the KGSL model performed consistently better than LightGCN on all three datasets, and showed slight improvements compared to the MCCLK model in the evaluation metrics. This demonstrates the effectiveness of the self-supervised learning task proposed in KGSL in assisting with the recommendation task. The self-supervised learning task of KGSL not only leverages the user–item interaction graph to find self-supervised signals, but also utilizes the semantic similarity in the item similarity graph to generate self-supervised signals. To some extent, it mitigates the instability resulting from relying on a single source for generating self-supervised signals. Therefore, it is possible to enhance model training and alleviate the impact of data sparsity by employing more effective and diverse self-supervised signals, ultimately improving the accuracy of recommendations.

### 5.3. Ablation Study of the KGSL Framework

In this section, to further validate the effect of the self-supervised learning task on KGSL performance, which is constructed by utilizing the semantic-based item similarity graph for data augmentation, we designed two variant models of the KGSL model in the Top-10 recommendation task, namely KGSL-NS and KGSL-NK. KGSL-NS is used to eliminate the data augmentation operation of the user–item interaction graph. It exclusively utilizes GNN for node representation operation and does not include self-supervised learning on semantic-based item similarity graphs. KGSL-NK is a without-data augmentation operation for the semantic-based item similarity graph and without a self-supervised signal

for the user–item interaction graph. The results of the ablation experiment are shown in Table 4.

From Table 4, we observe that KGSL-NK outperforms KGSL-NS on the three datasets. This demonstrates that employing self-supervised learning through data augmentation on its own data can generate more and more effective information. This provides additional self-supervised signals that alleviate data sparsity issues and enhance the performance of the recommendation model. In addition, KGSL demonstrates the best performance. This indicates that constructing self-supervised learning tasks based on item semantic similarity effectively enhances the quality of self-supervised signals. This, in turn, helps to avoid interference from noisy data in the model, resulting in more accurate and effective recommendation outcomes.

**Table 4.** Performance compared with model variants of KGSL.

| Dataset | Metrics | KGSL-NS | KGSL-NK | KGSL |
|---------|---------|---------|---------|------|
| MovieLens-1M | Recall@10 | 24.238 | 24.729 | 25.875 |
| | NDCG@10 | 22.074 | 22.738 | 23.834 |
| Last-FM | Recall@10 | 27.829 | 28.445 | 29.585 |
| | NDCG@10 | 20.586 | 21.163 | 22.181 |
| Book-Crossing | Recall@10 | 9.415 | 9.495 | 9.952 |
| | NDCG@10 | 5.884 | 6.122 | 6.174 |

*5.4. Hyperparameter Sensitivity Analysis*

In this section, extensive experiments were conducted on four key hyperparameters used in the KGSL model: the self-supervised learning scale weight coefficients $\beta$, the number of self-supervised signals $K$, the number of layers of the graph neural network encoder $L$, and the embedding dimension of the entities $d$.

Experiments were conducted with different settings of the self-supervised learning scale weight coefficients $\beta$ on the three datasets, and the results obtained are shown in Figure 5.

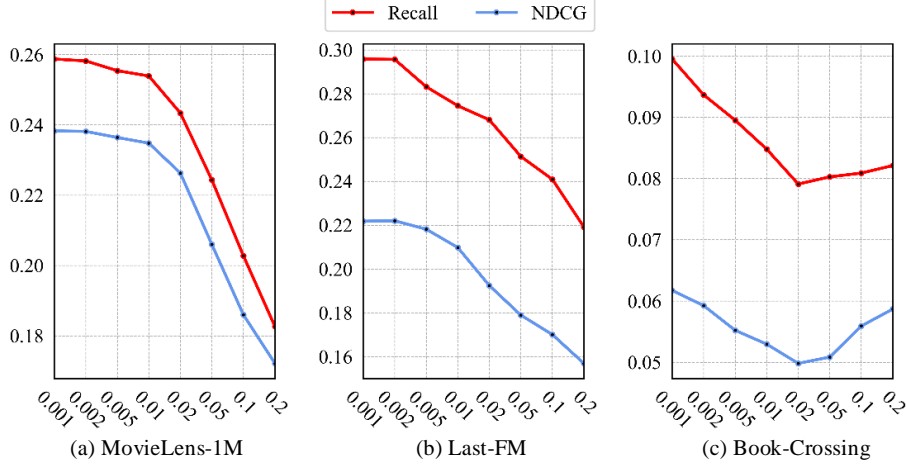

**Figure 5.** KGSL performance w.r.t. the SSL Scale Weight Coefficient $\beta$ on three datasets.

In analyzing the influence of $\beta$ on the model, $K = 30$ was set on MovieLens-1M and Last-FM, and $K = 40$ on Book-Crossing. From Figure 5, it can be observed that KGSL is highly sensitive to the value of $\beta$. When $\beta$ takes smaller values, the model achieves desirable performance, while larger values lead to performance degradation. In a comprehensive analysis, for the three datasets, the weight coefficients of controlling self-Supervised Learning tasks were set to $\beta = 0.001$.

The results obtained from the experiments conducted on the three datasets with different number of self-supervised signals $K$ are shown in Figure 6.

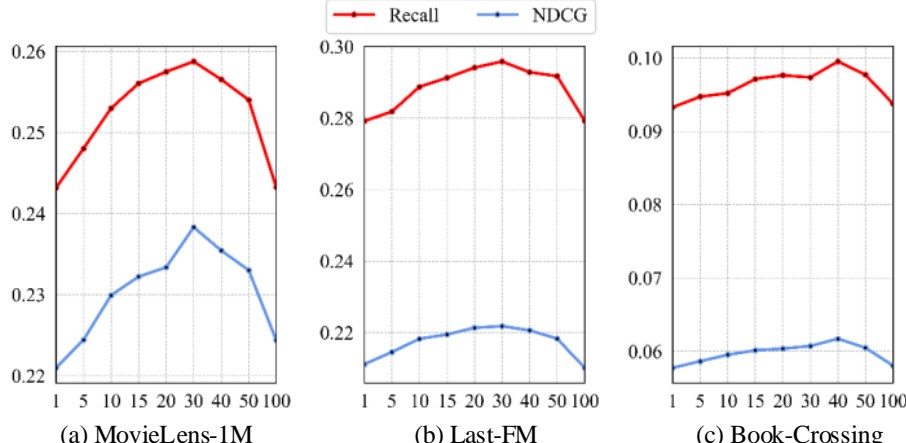

**Figure 6.** KGSL performance w.r.t. the number of self-supervised signals $K$ on three datasets.

In analyzing the effect of $K$ on KGSL, we set $\beta = 0.001$. As can be seen in Figure 6, as the number of self-supervised signal increases, the performance of KGSL improves slowly to the maximum and then declines on the Last-FM and Book-Crossing datasets. However, on the MovieLens-1M dataset, the KGSL performance improves faster to the highest point and then decreases. This phenomenon can be explained from the perspective of the dataset. When the number of items is relatively small, the impact of the self-supervised signals generated based on item semantic similarity on KGSL performance is more significant as the item quantity increases. This indicates that such an approach can provide high-quality self-supervised signals, effectively mitigating data sparsity. Combining the results from Figure 6 and the characteristics of the datasets, the number of self-supervised signals for MovieLens-1M, Last-FM, and Book-Crossing datasets is set to $K = 30$, $K = 30$, $K = 40$, repectively.

The results obtained from the experiments conducted on the three datasets with different numbers of graph encoder layers $L$ are shown in Figure 7.

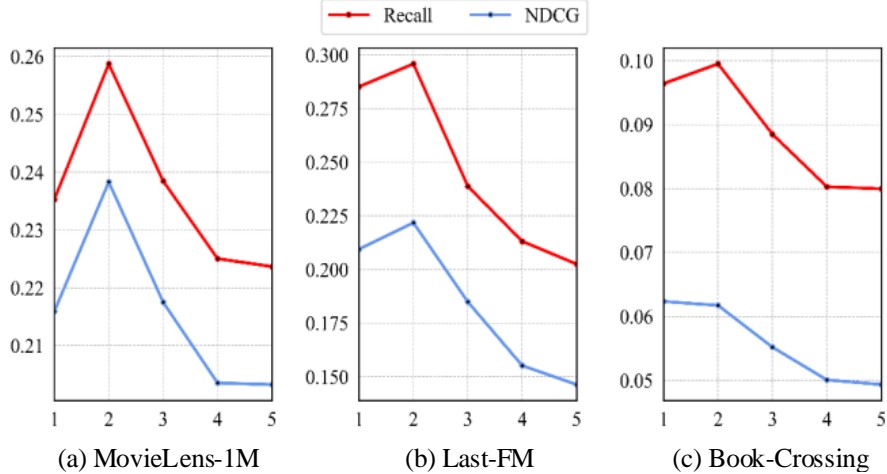

**Figure 7.** KGSL performance w.r.t. the Neural Network Encoder Layer $L$ on three datasets.

As seen in Figure 7, it can be observed that as the number of encoder layers increases, the NDCG values show a declining trend on the Book-Crossing dataset, while on MovieLens-1M and Last-FM, it initially increases and then decreases. The Recall value reaches its maximum at $L = 2$, and as $L$ increases further, there is a significant drop in model performance. This phenomenon suggests that longer relationship chains are not practically meaningful when inferring item similarity, and they can lead to lower-quality

self-supervised signals for the model. Therefore, the number of network layers was set $L = 2$ for all three datasets.

In the Top-10 recommendation task, we investigated the impact of entity embedding dimension $d$ on the KGSL model. The experimental results are shown in Figure 8. As $d$ increases, there is a corresponding increase in model performance, indicating that a higher dimension $d$ can encode more information between items and entities during knowledge representation learning. However, as $d$ continues to increase, model performance starts to decline. This suggests that an excessively high embedding dimension $d$ can lead to overfitting in the knowledge representation learning process, resulting in lower-quality entity embedding. This, in turn, affects the construction of the semantic-based item similarity graph, and prevents the self-supervised learning task from acting as an effective assistant to the main task. Based on the experimental results, we set the representation dimensions for the three datasets Movie-, Lens-M, Last-FM, and Book-Crossing as $d = 16$, $d = 32$, $d = 32$, respectively.

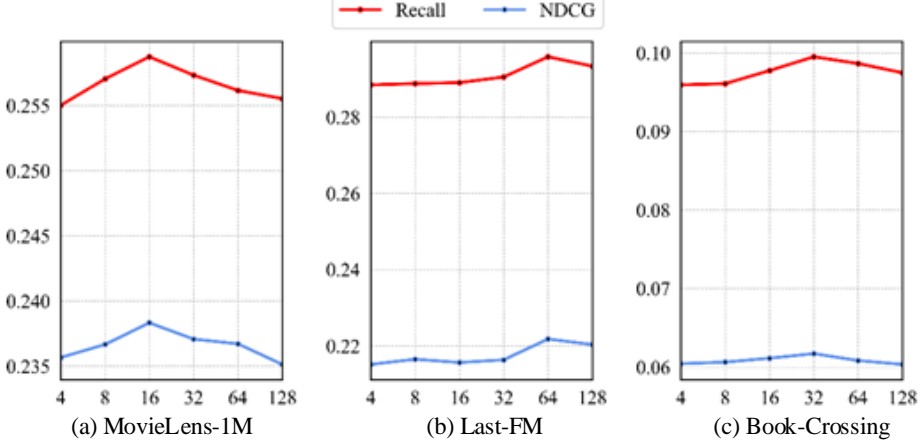

**Figure 8.** KGSL performance w.r.t. the entity embedding dimension $d$ on three datasets.

### 5.5. Study on the KGSL Effectiveness

In this section, the training process of KGSL is analyzed under the Top-10 recommendation task. The loss variation curves of KGSL training on the three datasets are shown in Figure 9, where the horizontal coordinate indicates the number of training iteration epochs and the vertical coordinate indicates training loss $L_{KGSL}$ of the model.

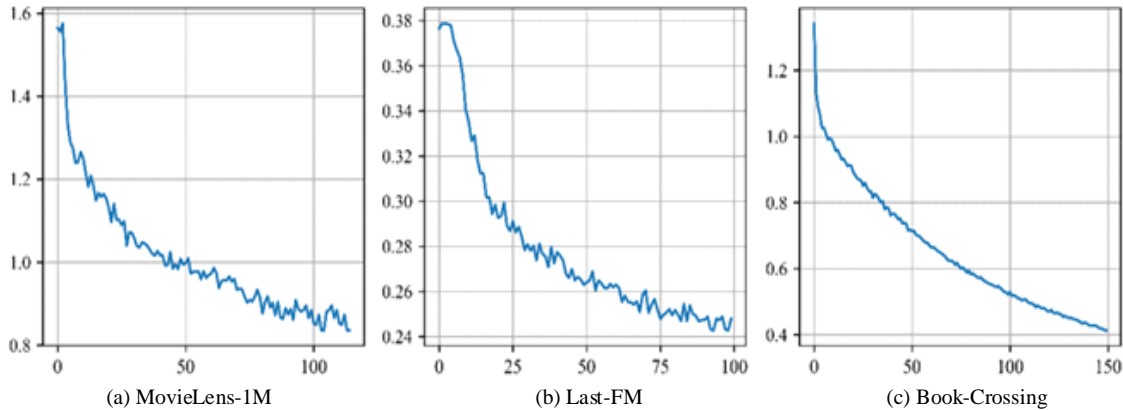

**Figure 9.** The loss cureve of KGSL.

Since the convergence speed of KGSL on the three datasets is inconsistent, the maximum training iteration numbers for KGSL on MovieLens-1M, Last-FM, and Book-Crossing are set to 120, 100, and 150, respectively. As seen from Figure 9, during the early stages of model training, the training loss decreases sharply on all three datasets. However, as

the number of training iterations increases, the training loss gradually levels off. This demonstrates that the KGSL model converges effectively and is relatively easy to train.

## 6. Conclusions

In this work, we analyzed the limitations of recommendation algorithms based on graph neural networks and proposed the Knowledge-Aware Graph Self-Supervised Learning (KGSL) algorithm. First, we constructed a semantic-based item similarity graph and designed data augmentation views. Then, we utilized LightGCN to learn node representations within the graph and created an unlabeled dataset. We designed a self-supervised learning task that combines structure and semantics information, providing additional self-supervised signals to the model. Finally, we introduced a joint optimization strategy that combines self-supervised and supervised learning, creating an end-to-end model. During the model training process, the KGSL model leverages self-supervised learning to predict extra self-supervised signals from raw data, aiding in the learning of node representations, and mitigating the challenges posed by data sparsity. In addition, we harnessed item embeddings containing diverse information from raw data to seek self-supervised signals. This enhances model ability to learn representations of items that users have not interacted with, which in turn better models user preferences and improves the accuracy of KGSL recommendations. Finally, extensive experiments were conducted to validate the model performance on three real and widely used recommendation datasets. The results demonstrate that KGSL has certain advantages compared to state-of-the-art models. Furthermore, we conducted ablation experiments to confirm the positive impact of the constructed self-supervised tasks on the model's recommendation performance.

**Author Contributions:** Study design and writing, S.L.; literature search, Y.J. and Y.W.; figure, N.W. and L.Z.; supervision, J.G. All authors have read and agreed to the published version of the manuscript.

**Funding:** This research was funded by the National Natural Science Foundation of China under Grant 42306218, the S&T Program of Hebei under Grant 226Z0102G, the Hebei Natural Science Foundation under Grant F2023407003, the S&T Plan Project of Hebei Academy of Sciences under Grant 2023PF01.

**Data Availability Statement:** All data generated or analyzed during this study are included in this published article.

**Conflicts of Interest:** The authors declare no conflict of interest.

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
