# Peer review of "Knowledge-Aware Graph Self-Supervised Learning for Recommendation"

_electronics, doi:10.3390/electronics12234869_

Round 1
Reviewer 1 Report
Comments and Suggestions for Authors
Reviewer 2 Report
Comments and Suggestions for Authors
This paper proposes the development of a recommendation system (KGSL) utilizing methods from the research areas of Graph Neural Networks (GNNs) and Self-Supervised Learning (SSL). The proposed approach exploits three different graphs (namely, User-Item Graph, Item Knowledge Graph, and Semantic-based Item Similarity Graphs) with a LightGCN network and a contrastive learning technique (namely, InfoNCE) to augment item embeddings and capitalize on structural and semantic information hidden in the graph representations.
The authors initially present a variety of recommendation systems that have been developed during the last years, emphasizing very clearly the advantages and the limitations of each perspective. Moreover, they elaborate on the related research approaches on GNN-based recommendation systems and SSL techniques employed to alleviate data sparsity. They provide substantial feedback regarding each approach, while clarifying the factors differentiating them. However, it is advisable that the authors include an introduction for both GNNs and SSL methods so that the reader can easily navigate through the rest of the related work. As far as the preliminaries are concerned, the authors should provide more details for the graphs they use. Specifically, they should comment on the nature and the characteristics of User-Item and Item Knowledge Graph. Nevertheless, Table 1, which incorporates all the mathematical notations employed throughout the rest of the paper is a nice addition.
The authors present the KGSL model in a well-structured manner, elaborating on every aspect and component of their approach. The analysis of the construction of Semantic-based Item Similarity Graph, the development of LightGCN and InfoNCE networks are comprehensive and thorough. However, we suggest that the authors provide more details about Figure 4, as its role to the construction of the Item Similarity Graph is not very translucent. The evaluation process is robust with a plethora of datasets utilized to showcase the state-of-the-art results the proposed model achieves. The baseline model they employ for the experiments are some of the most popular ones in the academia and are presented in an elaborated fashion. While the experiment results are easy to follow, the authors could have added the Hits@K metric, which accompanied with the Recall and NDCG@K metrics they have employed, are the standard metrics used to compare recommendation systems.
Their ablation study examines the impact of a series of hyperparameters to the KGSL model. Both the figures included in the paper and the accompanying analysis are exhaustive, thus ensuring the readers about the robustness of the model proposed.
Overall, the quality of the manuscript is excellent with minor alterations needed. Authors may also want to consider related works, such as:
* Sun, Y., Zhu, J., & Xi, H. (2022, September). Knowledge-Aware Self-supervised Graph Representation Learning for Recommendation. In International Conference on Artificial Neural Networks (pp. 420-432). Cham: Springer Nature Switzerland.
* Kanakaris, N., Giarelis, N., Siachos, I., & Karacapilidis, N. (2021). Shall I Work with Them? A Knowledge Graph-Based Approach for Predicting Future Research Collaborations. Entropy, 23(6), 664.
* Yang, Y., Huang, C., Xia, L., & Huang, C. (2023, August). Knowledge Graph Self-Supervised Rationalization for Recommendation. In Proceedings of the 29th ACM SIGKDD Conference on Knowledge Discovery and Data Mining (pp. 3046-3056).
* Kanakaris, N., Giarelis, N., Siachos, I., & Karacapilidis, N. (2022). Making personnel selection smarter through word embeddings: A graph-based approach. Machine Learning with Applications, 7, 100214.
As a final remark, the authors are advised to make their code publicly accessible, for reproducibility purposes.

Comments on the Quality of English LanguageThe level of the English is appropriate and the paper is highly understandable. The quality of the English language is adequate, with minor syntactical and lexical errors. The authors are advised to navigate through the manuscript and correct these mistakes.
Reviewer 3 Report
Comments and Suggestions for Authors
In this paper, the author propose a self-supervised method to jointly analyze the user-item graph and knowledge graph via contrastive learning. They exploit the node similarity to augment the graph. Moreover, they consider the semantic and structural information simultaneously in the contrastive framework. To evaluate the performance of their method, they compare the method with other baselines. The topic is quite innovative. I suggest the author to make a minor revision. After reading this paper, I have following suggestions:
1st The author developed a graph data augmentation based method. I suggest the author to explain the method to select the hard-negative samples since it is proved to be crucial for contrastive learning.
2nd As they consider to use both semantic and structural to improve the robustness, I suggest the author to clearly discuss whether the loss will be affected when there exist some counterfactual data.
3rd The related work need to be enriched. I suggest the author to introduce some recent application of graph neural application on recommendation system:
1 Wu, Shiwen, Fei Sun, Wentao Zhang, Xu Xie, and Bin Cui. "Graph neural networks in recommender systems: a survey." ACM Computing Surveys 55, no. 5 (2022): 1-37.
2 Gao, Chen, Xiang Wang, Xiangnan He, and Yong Li. "Graph neural networks for recommender system." In Proceedings of the Fifteenth ACM International Conference on Web Search and Data Mining, pp. 1623-1625. 2022.
3 Chen, Ziheng, Fabrizio Silvestri, Jia Wang, Yongfeng Zhang, Zhenhua Huang, Hongshik Ahn, and Gabriele Tolomei. "Grease: Generate factual and counterfactual explanations for gnn-based recommendations." arXiv preprint arXiv:2208.04222 (2022).
4 Boratto, Ludovico, Francesco Fabbri, Gianni Fenu, Mirko Marras, and Giacomo Medda. "Counterfactual Graph Augmentation for Consumer Unfairness Mitigation in Recommender Systems." In Proceedings of the 32nd ACM International Conference on Information and Knowledge Management, pp. 3753-3757. 2023.
5 Huang, Tinglin, Yuxiao Dong, Ming Ding, Zhen Yang, Wenzheng Feng, Xinyu Wang, and Jie Tang. "Mixgcf: An improved training method for graph neural network-based recommender systems." In Proceedings of the 27th ACM SIGKDD Conference on Knowledge Discovery & Data Mining, pp. 665-674. 2021.
6 Nguyen, Thanh Toan, Khang Nguyen Duc Quach, Thanh Tam Nguyen, Thanh Trung Huynh, Viet Hung Vu, P. Le Nguyen, Jun Jo, and Quoc Viet Hung Nguyen. "Poisoning GNN-based recommender systems with generative surrogate-based attacks." ACM Transactions on Information Systems (2022).
Comments on the Quality of English LanguageGenerally good, need to polish some sentences.
